# Profiling DNA damage response following mitotic perturbations

Ronni S. Pedersen[1], Gopal Karemore[1], Thorkell Gudjonsson[1,†], Maj-Britt Rask[1], Beate Neumann[2], Jean-Karim Hériché[3], Rainer Pepperkok[2,3], Jan Ellenberg[3], Daniel W. Gerlich[4], Jiri Lukas[1] & Claudia Lukas[1]

Genome integrity relies on precise coordination between DNA replication and chromosome segregation. Whereas replication stress attracted much attention, the consequences of mitotic perturbations for genome integrity are less understood. Here, we knockdown 47 validated mitotic regulators to show that a broad spectrum of mitotic errors correlates with increased DNA breakage in daughter cells. Unexpectedly, we find that only a subset of these correlations are functionally linked. We identify the genuine mitosis-born DNA damage events and sub-classify them according to penetrance of the observed phenotypes. To demonstrate the potential of this resource, we show that DNA breakage after cytokinesis failure is preceded by replication stress, which mounts during consecutive cell cycles and coincides with decreased proliferation. Together, our results provide a resource to gauge the magnitude and dynamics of DNA breakage associated with mitotic aberrations and suggest that replication stress might limit propagation of cells with abnormal karyotypes.

[1] Protein Signaling Program, Novo Nordisk Foundation Center for Protein Research, Faculty of Health and Medical Sciences, University of Copenhagen, Blegdamsvej 3B, DK-2200 Copenhagen, Denmark. [2] Advanced Light Microscopy Facility, European Molecular Biology Laboratory, Meyerhofstr. 1, 69117 Heidelberg, Germany. [3] Cell Biology and Biophysics Unit, European Molecular Biology Laboratory, Meyerhofstr. 1, 69117 Heidelberg, Germany. [4] Institute of Molecular Biotechnology of the Austrian Academy of Sciences (IMBA), Vienna Biocenter (VBC), Dr Bohr-Gasse 3, 1030 Vienna, Austria. † Present address: Department of Biochemistry and Molecular Biology, Faculty of Medicine, University of Iceland, 101 Reykjavik, Iceland. Correspondence and requests for materials should be addressed to J.L. (email: jiri.lukas@cpr.ku.dk) or to C.L. (email: claudia.lukas@cpr.ku.dk).

Proliferating cells are constantly challenged by endogenous DNA damage including the most destructive DNA double-strand breaks (DSBs)[1]. This poses a challenge for genome surveillance because even sporadic DSBs can destabilize the genome[2]. Among the main sources of endogenous DSBs are errors during DNA replication, so called replication stress (RS)[3–5]. Interestingly, RS-coupled DSBs are rarely generated during S phase due to the surplus of replication protein A, which shields replication intermediates against nucleolytic attacks[6]. The prevailing mode by which RS destabilizes the genome is by creating substrates that are converted to DNA breaks only during mitosis[7]. Amongst prominent examples of this trait are common fragile sites, which due to paucity of replication origins and topological constraints that obstruct movement of replication forks fail to complete DNA replication in one cell cycle[8]. This generates DNA structures that cannot be detected by cell cycle checkpoints and are therefore transferred to mitosis where they become converted to DSBs via the MUS81-EME1 nuclease[9]. Hence, RS and mitosis are intrinsically coupled by 'trading' DNA breakage (which can be repaired) for the possibility to complete chromosome segregation (whose failure would be lethal). Although many RS-initiated and mitosis-executed DSBs can be repaired already during mitosis[10], a fraction of these lesions is frequently transferred to daughter cells where they become sequestered in 53BP1 nuclear bodies until they are repaired[11,12]. Should any of these mechanisms fail, RS-induced and mitosis-propagated DSBs can give rise to structural and numerical chromosome instability, which could in turn fuel cancer progression[13].

Besides this role of otherwise normal mitosis in processing RS intermediates, primary mitotic errors also seem to contribute to the acquisition of DNA breakage[14,15]. For instance, it was reported that DNA trapped in the cytokinesis furrow might break and thereby generate templates for chromosomal translocations[16]. In addition, daughter cells connected by dicentric chromosomes can acquire DSBs through nuclear envelope rupture, which exposes chromosomes to cytosolic nucleases[17]. Furthermore, cells forced to undergo long mitotic delays by microtubule poisons may develop DSBs via exhaustion of the telomere-protecting shelterin complex[18].

Finally, chromosome missegregation can also undermine genome integrity by triggering numerical chromosome abnormalities[19]. However, genomes of polyploid and aneuploid cells tend to be unstable and develop DNA damage later in their life span[14,20]. How that happens has long been unknown, until recently, when several studies showed that chromosome missegregation is accompanied by hallmarks of RS. Most notably, it was reported that DNA replication in micronuclei proceeds in an untimely and erratic fashion and leads to DNA damage[21,22]. Another study showed that genome instability in aneuploid cells is associated with reduced expression of the minichromosome maintenance (MCM) replicative helicase, again pointing to RS as a source of DNA damage after impaired chromosome segregation[23]. Although intriguing, the generality of this hypothesis has not been tested and it is currently unknown whether other types of mitotic aberrations can also impair DNA replication. Even more importantly, it remains unclear how mitotic errors generate DSBs and what (if any) is the role of RS in cells confronted with mitotic perturbations. To shed light on these issues and generate resource for their further investigation, we systematically silence by siRNA a representative set of established cell cycle regulators, whose disruption impairs major mitotic events[24]. By combining this approach with multiparametric profiling of the cell population data, and together with real-time tracking of single cells for several successive generations, we ask whether mitotic errors and DNA

breakage in daughter cells are functionally connected, whether RS is involved, and how mitosis-induced DNA-damage response (DDR) affects cell fate decisions.

## Results

**Conditions to study crosstalk between mitotic errors and DDR.** As a cellular model we used U-2-OS, a human osteosarcoma cell line that has been extensively characterized for DDR including sporadic DSBs generated during cell cycle progression[12]. U-2-OS cells have other advantages due to favourable morphology for automated microscopy, high efficiency of RNAi and availability of isogenic derivatives stably expressing fluorescently tagged proteins that allow monitoring of both mitotic and DDR events (see Methods). In addition, U-2-OS cells harbour one functional allele of p53 (ref. 25) and express elevated level of MDM2, a general suppressor of p53 (ref. 26). This limited but not completely absent p53 response allowed us to add p53 to the list of mitotic stress readouts, and at the same time exploit the partially compromised G1 checkpoint in U-2-OS to monitor consequences of mitotic errors in successive cell generations. Such attenuated p53 response resembles early stages in oncogenic transformation, where the incipient tumour cells undergo clonal selection and where errors during chromosome segregation can play a decisive role in directing cell fate events.

To interfere with mitosis at main transitions, we mined the Mitocheck database, which harbours over 1,000 potential regulators of cell division identified in HeLa cell line by RNA interference[24]. We re-tested all Mitocheck siRNAs for their ability to induce matching mitotic phenotypes in U-2-OS cells and subjected all positive hits to additional rounds of validation to remove targets for which two siRNAs produced divergent results due to potentially confounding RNAi seed effects[27]. As the result, we obtained 47 validated siRNA pairs (two independent siRNAs for each target), which consistently produced expected mitotic phenotypes, and which thus provided us with a high-confidence molecular tool to perturb mitosis in a defined and stringently controlled fashion (Supplementary Fig. 1; Supplementary Dataset 1).

To assess the impact of the above described mitotic perturbations on DDR, we applied multiparametric profiling to quantify mitotic phenotypes (mitotic figures, morphological aberrations of DAPI-stained nuclei), DNA replication (incorporation of the nucleotide analogue EdU), general cellular stress (accumulation of p53) and two independent markers of DDR, γ-H2AX and 53BP1 respectively. γ-H2AX is one of the earliest DDR responses, which has been widely used to monitor a variety of DNA damage including DSBs[28]. 53BP1 is a chromatin-bound DDR mediator, which complements γ-H2AX by its ability to report low or transient DNA breakage owing to its long residence time at DSB sites[29]. In anticipation of complex phenotypes, we analysed quantitative image features by logistic regression (LR), a supervised machine-learning tool designed to integrate multiple image features and generate probability scores of their frequency and magnitude at the level of single cells. Schematic workflow of the phenotypic profiling including validation steps is depicted in Supplementary Fig. 2a,b and further detailed in Methods.

**Mitotic aberrations increase probability of DSBs.** Having established the experimental conditions, we knocked down all 47 mitotic regulators and applied LR profiling to assess the ensuing cellular responses including 53BP1 accumulation at DSBs. We observed a clear trend whereby an increased occurrence of mitotic aberrations correlated with a higher probability

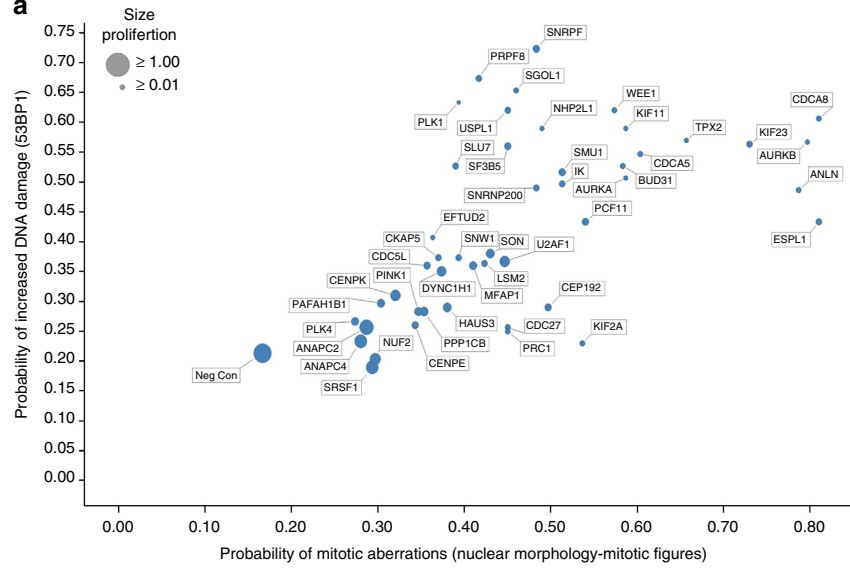

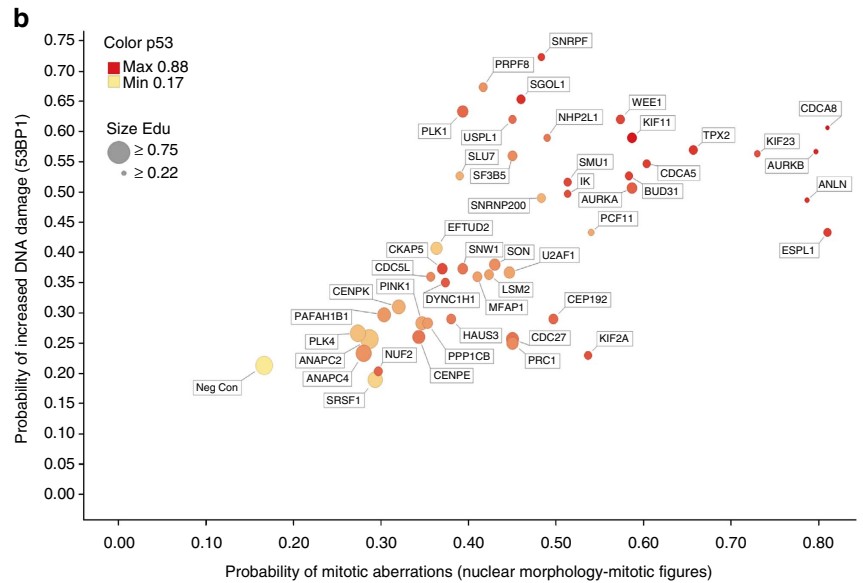

**Figure 1 | Probabilistic profiling of DDR and associated phenotypes upon mitotic perturbations.** (**a**) Probability scores were generated by LR and are displayed to show correlations between mitotic phenotypes and 53BP1-decorated DNA lesions. Values reflect scores derived from $n = 3$ independent siRNA transfections. Each target is additionally classified according to cell proliferation (dot size is proportionate to relative cell count at the time of evaluation). Data for one of the two siRNAs (siRNA #1) for each depicted target are displayed. Numerical values for both siRNAs and all readouts are provided in Supplementary Dataset 2. (**b**) The same probability plot as in **a** where each target is additionally classified according to DNA replication (decreasing dot size reflects reduction of EdU incorporation) and p53 accumulation (heat map reflects p53 protein levels). The EdU data were derived form the same experiment as in **a**. Probability scores for p53 were derived from a separate experiment performed in parallel under identical conditions in $n = 3$ independent siRNA transfections.

of DNA breakage in daughter cells (Fig. 1a). This correlation was reproduced by γ-H2AX as an alternative DSB marker (Supplementary Fig. 3a,b). The DNA damage markers and mitotic error phenotypes were also correlated with reduced cell proliferation (Fig. 1a, dot size), less EdU incorporation (Fig. 1b, dot size) and higher levels of p53 (Fig. 1b, dot colour). Despite this clear trend in phenotype correlations, we noted that some RNAi conditions (for example, NUF2, HAUS3, CEP192) caused mitotic errors, stabilization of p53 and reduced proliferation without a substantial increase in DNA breakage (Fig. 1b). We found these observations intriguing in light of the experimental conditions under which they were acquired. On the one hand, our results were in line with the mounting

evidence that p53 can sense mitotic errors independent of DDR[30,31]. On the other hand, the strong correlation between increased mitotic errors and decreased proliferation at the cell population level was unexpected given the attenuated G1 checkpoint in U-2-OS cells (see Introduction) and suggested existence of a functional intermediate between primary mitotic errors and cell fate decisions that is independent of p53. We will return to, and further elaborate on, this hypothesis throughout the study. From the validation point of view, profiling of all main phenotypes depicted in Fig. 1 was reproduced using independent siRNAs to all 47 interrogated genes (Supplementary Fig. 2b). A comprehensive summary of all these data derived from large cell numbers (average 700 cells

per siRNA) is provided in Supplementary Dataset 2, and supported by rigorous statistics in Supplementary Dataset 3. These data sets are the essentials of the resource to gauge the magnitude and penetrance of DNA breakage associated with major types of mitotic aberrations.

**Perturbed mRNA splicing triggers DSBs independent of mitosis.** The above experiments clearly indicated that mitotic errors are accompanied by an increased probability of DNA breakage, yet they did not reveal whether these phenotypes are functionally coupled. To address this, we individually knocked down each of the 47 mitotic genes in two cell lines derived from the same parental U-2-OS strain, one expressing fluorescently tagged histone H2B (to monitor nuclear aberrations caused by mitotic errors), and the other one co-expressing 53BP1 (to monitor DDR) with PCNA (to assess DNA replication). We then subjected these cells to parallel time-lapse microscopy and recorded long-term movies to determine the sequential order of mitotic errors and DNA damage in the newly born daughter cells. Strikingly, we found that while many knockdowns caused mitotic abnormalities (Fig. 2a) or mitotic delays (Fig. 2b) before 53BP1 accumulation at sites of DNA damage, knockdown of a smaller group of genes enriched in mRNA splicing factors

caused mitotic aberrations only after DNA damage became clearly detectable (Fig. 2a). This phenotype order was validated by visual inspection of individual cell trajectories, which for example revealed that after knocking down PRPF8, a fraction of cells accumulated high DNA damage during S phase and into G2, but such cells typically did not enter mitosis and often died (Supplementary Fig. 4a, top). The other fraction of cells in the same RNAi-treated population entered mitosis without detectable DNA damage. Such cells then showed pronounced mitotic delays and segregation problems, which either resulted in intra-mitotic cell death (Supplementary Fig. 4a, middle) or in some cases in mitotic exit, which was however not accompanied by a massive increase of DNA breakage in G1 daughter cells (Supplementary Fig. 4a, bottom). The decision between these cell fates seems stochastic and likely depends on which rate-limiting factor becomes depleted earlier. Taken together, these data suggest that mitotic errors and DNA damage are mechanistically separate outcomes of perturbed mRNA splicing. We regard this result as an important extension of this resource because mRNA splicers frequently score in DNA damage and cell cycle screens[24,32–34], yet their mutual relationship has thus far remained elusive.

Interestingly, one gene that is not directly involved in mRNA metabolism but whose knockdown showed a similar temporal

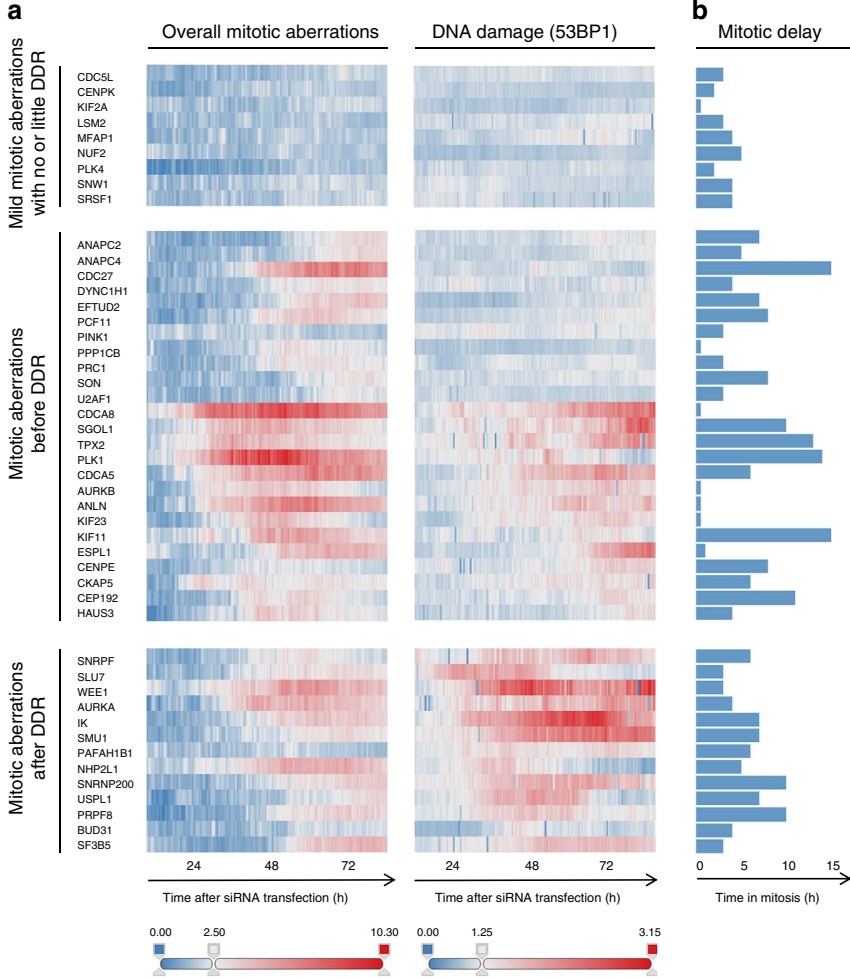

**Figure 2 | Temporal profiling of mitotic aberrations and DDR.** (**a**) Heat maps summarizing data from representative independent time-lapse recordings ($n = 3$) of U-2-OS cell lines stably expressing mCherry-H2B for mitotic abnormalities or mRFP-53BP1 for DNA damage after transfection with siRNAs to the indicated mitotic regulators. The phenotypes are sub-divided into three distinct temporal groups whose main characteristics are indicated. The heat maps are derived from at least 150 cells for each condition and represent fold increase values compared with mock-transfected cells. (**b**) Bar charts were derived from the same experiment as in **a** and depict average mitotic delay for each condition.

profile (DDR preceding mitotic phenotypes) was WEE1 (Fig. 2a). Also in this case we could confirm by visual inspection of single-cell trajectories that DNA damage and mitotic perturbation were largely uncoupled and readily occurred independently of each other, spanning interphase DNA damaged followed by extended G2 arrest (Supplementary Fig. 4b, top), mitotic cell death (Supplementary Fig. 4b, middle), and cell death after mitotic exit without intervening DDR (Supplementary Fig. 4b, bottom). Like for the mRNA processing factors, the uncoupling of interphase DDR and mitotic defects caused by WEE1 depletion might be explained by its biology: WEE1 kinase is an established suppressor of both interphase and mitotic CDK activities, and it has been well documented that its depletion followed by abrogation of cell cycle checkpoints generates major disruptions of both replication and mitosis[35,36], which as we show here are largely autonomous. As such, finding WEE1 within this temporal group was reassuring and supportive for the specificity of our real-time readouts.

**DSBs reflect biological penetrance of mitotic perturbations.** We next focused on gene perturbations characterized by mitotic defects preceding DDR. To start, we excluded all target genes representing splicing factors and WEE1 from the analysis as their depletion induces DDR indirectly and largely independent of mitosis (see the previous section). We then analysed the remaining genes by k-means clustering, a statistical tool for unbiased grouping of phenotypes (Fig. 3a, left). Using this approach, we derived four phenotypic clusters that represented distinct quantitative feature profiles (Fig. 3a, right). Whereas clusters 1 and 2 comprised mitotic aberrations accompanied

by low frequency of DNA damage, clusters 3 and 4 combined strong mitotic and DDR phenotypes (Fig. 3b). We thus hypothesized that the evolution of DDR in response to mitotic errors might reflect different penetrance of mitotic perturbations and tested this by visual inspection of individual cell trajectories for three phenotype-matched gene pairs that despite featuring similar mitotic aberrations appeared in distinct clusters in our assays.

We started by comparing two general mitotic regulators: CDC27 (cluster 2), a core subunit of the APC (Cyclosome), and PLK1 (cluster 3), an important mitotic kinase. Depletion of either protein caused mitotic arrest and the formation of DSBs marked by γ-H2AX. Strikingly however, the occurrence of DNA breakage was delayed and its magnitude was substantially less pronounced in CDC27-depleted cells (Fig. 4a) compared with the massive and early-onset DSB formation in PLK1-depleted cells (Fig. 4b). Both proteins were efficiently depleted with a very similar kinetics (Supplementary Fig. 5, and see also Supplementary Fig. 6 for original images of WBs), suggesting that DDR is not caused by mitotic delay *per se*, but reflects different functions of CDC7 and PLK1.

Notably, DSBs in CDC27- or PLK1-depleted cells strongly correlated with a permanent mitotic arrest. However, a much more frequent phenotype observed throughout the spectrum of interrogated genes was mitotic slippage[37,38], a mitotic delay of variable duration from which a subset of the affected cells eventually escaped and gave rise to daughter cells. We were therefore interested to test how cells tolerate transient mitotic delays, not least because unlike mitotic demise, the ability to proliferate in spite of mitotic errors is more pertinent to genomic instability that can cause disease. We first followed by time-lapse microscopy mRFP-53BP1-expressing cells depleted of

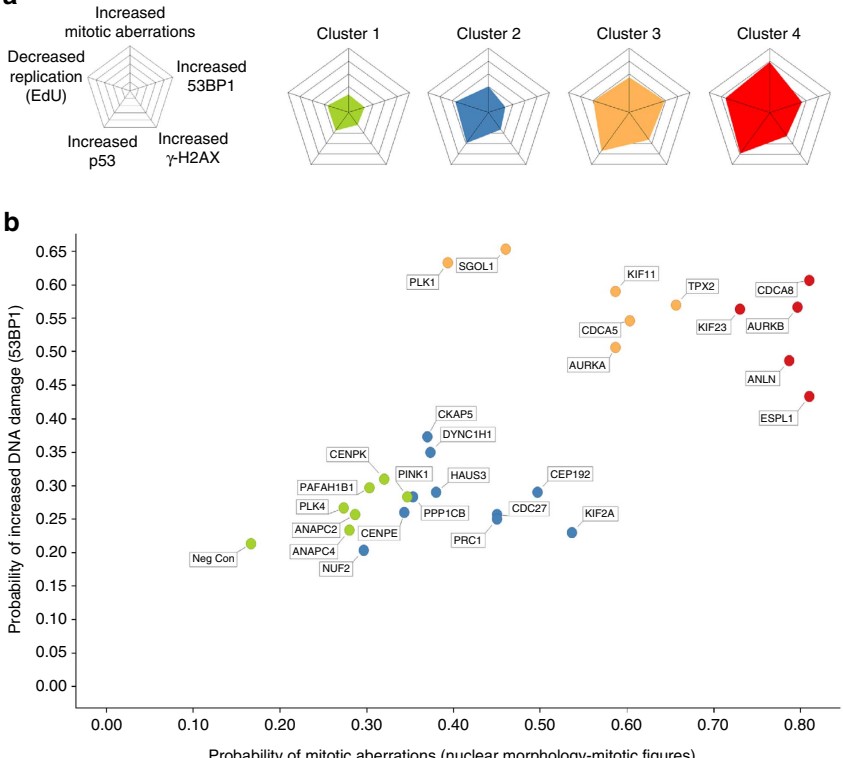

**Figure 3 | Identification of phenotypic similarity groups by computation-based clustering.** (**a**) Probability values for the five indicated phenotypes were subjected to k-means clustering (left) to sub-classify all siRNA targets into four clusters characterized by shared similarities (right). (**b**) The colour codes for each cluster identified in **a** are projected to the binary correlation between mitotic aberrations and DNA breakage derived from Fig. 1 after subtraction of splicing factors and WEE1 that induce DDR independent of mitotic perturbations.

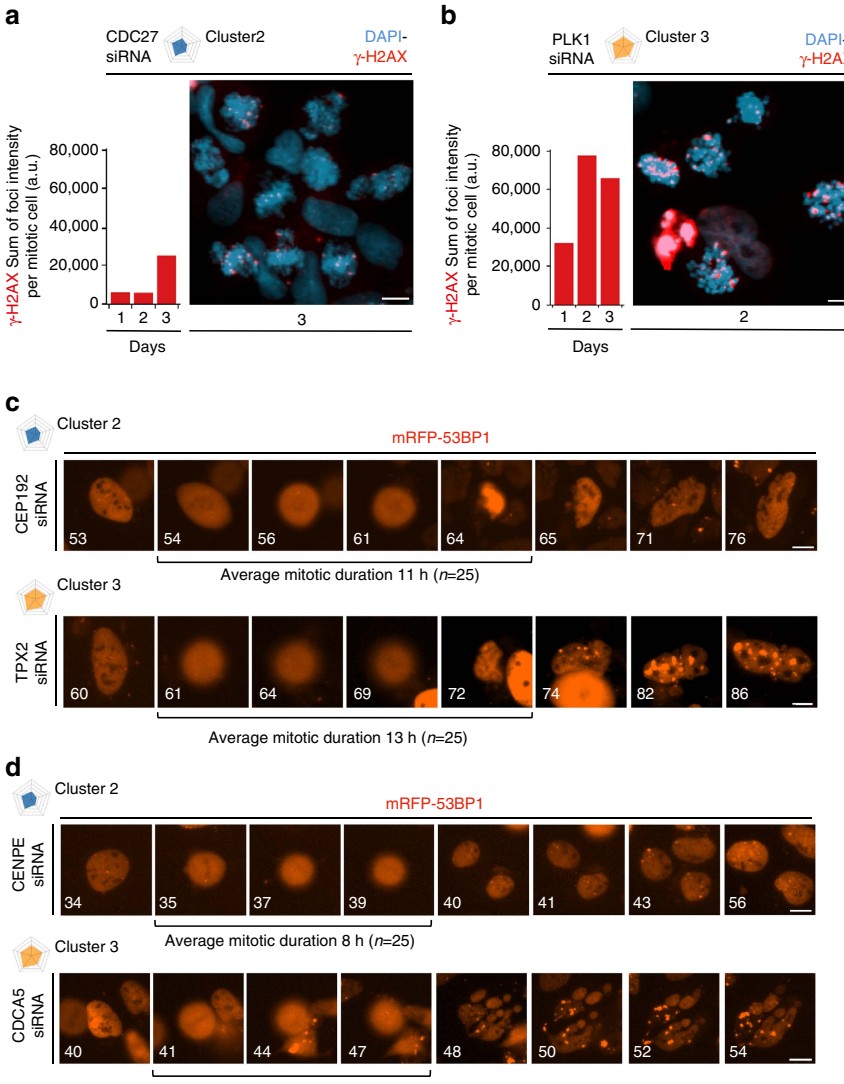

**Figure 4 | Biological penetrance of mitotic perturbation determines the timing and magnitude of mitosis-born DNA breakage.** (**a**) A representative image ($n = 50$) of U-2-OS cells treated with siRNA to CDC27 and immunostained for γ-H2AX. Nuclear DNA was counterstained with DAPI. Scale bar, 10 μm. The bar charts (left) show γ-H2AX-decorated DSBs in mitotic cells ($n = 250$) in the indicated times after siRNA transfection. (**b**) U-2-OS cells were treated with siRNA to PLK1 and analysed as in **a**. Scale bar, 10 μm. (**c**) Representative image galleries ($n = 25$) of U-2-OS cells expressing mRFP-53BP1, treated with the indicated siRNAs, and analysed by time-lapse microscopy. Average mitotic duration ($n = 25$) for each siRNA treatment is indicated. Numbers indicate time (h) after siRNA transfection. Scale bars, 10 μm. (**d**) Representative image galleries ($n = 25$) of U-2-OS cells expressing mRFP-53BP1, treated with the indicated siRNAs, and analysed as in **c**. Average mitotic duration ($n = 25$) for each siRNA treatment is indicated. Numbers indicate time (h) after siRNA transfection. Scale bars, 10 μm.

CEP192 (cluster 2) or TPX2 (cluster 3) regulators of mitotic spindle, each showing a high probability of mitotic phenotypes. Depletion of either factor caused severe mitotic spindle defects accompanied by several hours of mitotic delay and followed by a slippage of cells out of mitosis. However, we observed that despite a similar mitotic delay, cells deprived from genes classified in cluster 2 (CEP192; Fig. 4c, top) exited mitosis with much less DNA damage compared with cluster 3 (TPX2; Fig. 4c, bottom). Strikingly, we observed a very similar dichotomy after depletion of CENPE and CDCA5(Sororin), respectively. Also in this case, CENPE-depleted cells (cluster 2; Fig. 4d, top) had markedly less DNA breakage compared with the CDCA5(Sororin)-depleted cells (cluster 3; Fig. 4d, bottom) despite a very similar duration of mitotic delay. Because the RNAi efficiency was again very comparable for both proteins (Supplementary Fig. 5, and see also Supplementary Fig. 6 for original images of WBs), we propose that our results

might reflect what we operationally define here as 'biological penetrance'. Specifically, recent literature suggests that in addition to their established roles in mitosis, PLK1, TPX2 and CDCA5 (the cluster 3 members and stronger inducers of DDR in the above experiments) may have additional functions in interphase that can directly or indirectly impinge on genome integrity[39–43]. We therefore speculate that the mitotic chromosomes are in fact rather resilient and do not break readily when mitotic progression is halted. Rather, our findings suggest that the high incidence of mitosis-born DDR might be a combination of mitotic delay and chromosome fragility acquired in the preceding interphase. Viewed from such a perspective, our resource could guide future functional studies by indicating that mitotic regulators classified in clusters with pronounced DDR might have additional and hitherto concealed functions in genome integrity maintenance that may not be restricted to mitosis.

**Failed cytokinesis triggers DSBs via replication stress.** Finally, we were intrigued by the recurrent observation that in a number of instances when cells survived mitotic perturbations and gave rise to viable daughter cells, the first detectable signs of DNA breakage developed with a delay, often approximating an average duration of one cell cycle in U-2-OS (Fig. 2a). The temporal uncoupling between mitotic errors and the ensuing DDR was most discernible in cells that failed cytokinesis, which congregated in cluster 4 (Fig. 3). Cytokinesis failure leads to acute tetraploidy and progressive polyploidy over successive cell divisions that together define strong nuclear morphology aberrations and thus feature high on our probability plots. We were particularly intrigued by the notion that cytokinesis failure gives rise to binucleated cells (schematically depicted in Fig. 5a). Because the two daughter nuclei under such settings have normal morphology, we hypothesised that one source of DDR might be RS. Indeed, we noticed that depletion of the essential cytokinesis regulator ANLN progressively increased both number and intensity of 53BP1 nuclear bodies in G1 nuclei, which are very characteristic for RS generated at difficult-to-replicate chromosome fragile sites (Fig. 5b). We thus set out to test whether RS couples cytokinesis failure and DNA breakage after depleting the key regulators of this process (see Supplementary Figs 5 and 6 for siRNA efficiency).

To start, we inspected single-cell trajectories from time-lapse movies in U-2-OS cells expressing fluorescently tagged PCNA after depleting KIF23. We observed that following cytokinesis failure, the two nuclei that resided in a common cytoplasm developed multiple hallmarks of RS starting with slower rate of DNA replication. For instance, while the average S phase in control cells lasted 16 h, after depleting KIF23 it extended to 18 h and 22 h after the first and second aberrant mitosis, respectively. In some cases, the daughter nuclei re-replicated (Fig. 5c), and the replication between them was not synchronized and shifted for up to 6 h (Fig. 5d, top). In extreme cases, this resulted in cell division while one of the two nuclei was still replicating (Fig. 5d, bottom). This aberrant DNA replication was accompanied by DSBs as detected by 53BP1 (Fig. 5e) and $\gamma$-H2AX (Fig. 5f) that developed into an asymmetric pattern mirroring the unscheduled replication triggered by cytokinesis errors. Notably, labelling of newly replicated DNA by BrdU allowed us to capture events where DNA breakage clearly localized to active replicons (Fig. 5g).

Interestingly, RS and the ensuing DDR also occurred in cells that formed micronuclei due to nuclear fragmentation previously described as 'grape phenotype'[24] (Fig. 5h). DNA breakage in micronuclei strongly manifested as elevated $\gamma$-H2AX but only rarely as 53BP1 foci. To explain this dichotomy, we considered recent findings showing that RS and the ensuing DNA damage can be amplified by increased fragility of the nuclear membrane formed around missegregated chromosomes, which can lead to loss of important genome caretakers including 53BP1 (ref. 21). Consistent with this, we frequently observed disruptions of nuclear lamina accompanied by leakage of a soluble fraction of Gam-GFP DSB reporter[44]. Of note, a fraction of Gam-GFP remained associated with $\gamma$-H2AX foci, further supporting the presence of bona-fide DNA breakage in the fraction of fragmented nuclei that featured dysfunctional nuclear membrane (Fig. 5i). Together, these data indicate that micronucleation resulting from various RNAi conditions that perturb chromosome segregation leads to DNA breakage, which might be mediated by RS in micronuclei that lost replication or repair factors due to dysfunctional nuclear membrane. Of note, 53BP1 is an important mediator of DSB repair[45] and its loss from fragmented

nuclei may further exacerbate the RS-induced DNA breakage. In line with this, we recurrently observed grape phenotypes associated with permanent cell cycle arrest or cell death in S and G2 already after the first failed mitosis, whereas other types of mitosis-born, and RS-executed, DDR events allowed several consecutive cell cycles before proliferation ultimately ceased. In summary, we conclude that a variety of mitotic perturbations cause RS ranging from mild chromosome fragility in isolated loci to more general pan-nuclear stress.

## Discussion

We have systematically investigated how primary mitotic defects relate to genome integrity in consecutive cell generations. In our view, the key benefit provided by our resource and the analytical approaches therein is the deployment of LR machine-learning tools. This was instrumental to obtain robust and statistically significant probability scores and to generate an unbiased basis for phenotypic clustering. As such, these data should serve as a rich resource for future studies. Indeed, our results condensed to Fig. 5 aim at providing a proof of principle that this data set can be used to uncover novel aspects of a crosstalk between two major sources of intrinsic genome instability, namely chromosome portioning in mitosis and DNA replication during S phase. In the following paragraphs we elaborate on this latter aspect of our study with an emphasis on conceptual implications for diseases marked by unstable genomes such as cancer.

The key direction for functional exploration of the screening data was provided by the experimental setup itself, namely by the emphasis on multiparameter phenotypic profiling. Thanks to this approach, we noticed the unexpected temporal delay between binucleation and DNA breakage, and then continued to show that the latter is mediated by RS. These findings align well with an earlier study reporting RS as a cellular response to single-chromosome aneuploidy[23]. When combined with the systems approach undertaken in this study, we speculate that mitosis-born genome instability and RS are not only connected but that they are in fact inseparable. Earlier work showed that underreplicated CSFs are converted to DSBs during otherwise normal mitosis[9–12,46,47] and another study provided compelling evidence that RS-mediated DNA damage can trigger cancer-associated structural and numerical chromosome instability[13]. We find that the inverse scenario is also true and that primary mitotic errors are readily converted to DSBs via RS. Our data also indicate that mitotic errors as a source of RS-mediated DNA breakage are not restricted to structural chromosomal aberrations acquired during perturbed mitotic progression. Most notably in this regard, even two morphologically intact nuclei in a common cytoplasm developed progressive signs of RS after failed cytokinesis, suggesting that RS might be an obligatory outcome of mitotic perturbation. If correct, such assertion poses the key question of what is the biological role of RS triggered by primary mitotic errors, and if there is such role, then under which (pato)physiological settings it may become pertinent. Although we can only speculate at this point, there are intriguing analogies indicating that RS is not just a 'passive bystander' of mitotic failure. Compelling evidence has been provided that mitotic errors result in activation of the p53 tumour suppressor pathway. This counteracts further proliferation and thus places p53 as the first-in-line defence mechanism against propagating mitosis-born genome instability[31]. However, recent study showed that p53-mediated G1 arrest can be overcome by hyperactivation of growth factor signalling, an obligatory step during oncogenic transformation[30]. This suggests that cells might benefit from additional mechanisms that restrain propagation

of genome-destabilizing consequences of mitotic errors. We propose that RS can fulfil such a role because it triggers (and the ensuing DNA breakage amplifies) signalling cascades that halt cell cycle progression. This model has analogy with oncogenic stress, where RS functions as an inducible barrier against cancer progression[48,49]. However, and keeping with the analogy with

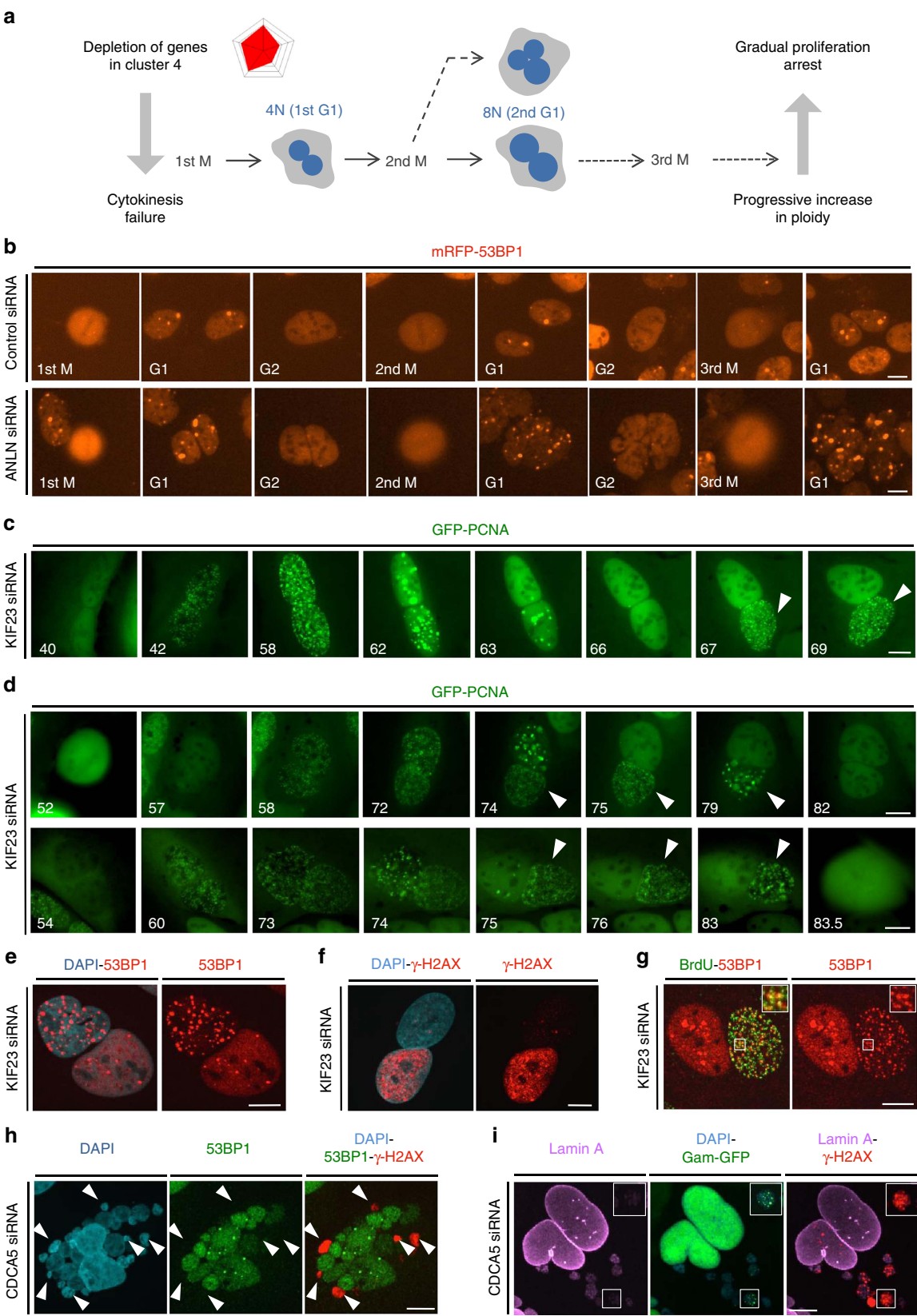

oncogene transformation, such proliferation barrier could be a double-edged sword because the cell cycle block in both scenarios generates a selective pressure to resume proliferation through mutating or silencing tumour suppressors. Indeed, tetraploid cells are known for their high susceptibility to oncogenic transformation[50] and our results suggest that this might be mediated, at least in part, by RS.

## Methods

**Cell lines and plasmids.** The human osteosarcoma cell line U-2-OS (authenticated by STR profiling, IdentiCell molecular diagnostics, Aarhus, Denmark) was cultured in DMEM medium (high glucose, Glutamax, Thermo Fisher Scientific, #31966-021) containing 10% of fetal bovine serum (FBS) and antibiotics (Penicillin, Streptomycin). $CO_2$-independent culture medium for live cell imaging (Thermo Fisher Scientific, #18045-054; containing Glucose, Glutamax, and Pyruvate concentrations as in DMEM, but without Phenol Red and Riboflavin) was custom-made and supplemented with 10% FBS and antibiotics. Several derivatives of the U-2-OS cell line stably expressing fluorescent protein-tagged genes of interest were generated by transient transfection with plasmids encoding resistance markers for Puromycin or Geneticin and isolation of colonies from single-cell dilutions ∼12 days after transfection. Stable cell lines expressing moderate levels of Histone H4-mEOS2, GFP-PCNA together with mRFP-53BP1, Histone H2B-mCherry and pGAM-GFP respectively, were characterized for full-length expression of the transgenes and correct protein localisation in the cells. Mammalian pEGFP-PCNA expression plasmid was a gift from Roland Kanaar (Erasmus University Medical Center, Rotterdam). Mammalian expression plasmid for GamEMGFP was a gift from Susan Rosenberg (Baylor Colleague of Medicine, Houston). pRSETa mEOS2 was a gift from Loren Looger (Addgene plasmid #20341) and a mammalian expression plasmid for Histone H4 was purchased from Origene (pCMV6-Myc-DDK-HIST4H4). All cell lines were regularly tested for Mycoplasma infection (MycoAlert, Lonza) and all tests were negative.

**siRNAs and transfection.** All siRNAs used in this study were obtained from Ambion—Thermo Fisher Scientific as *Silencer Select* reagents and used at a final concentration of 5 nM. Please see Supplementary Dataset 1 for siRNA IDs and sequences. All siRNA sequences were blasted for unique target specificity against human genome annotation data in ENSEMBL V75, 2014 (EMBL—EBI and Wellcome Trust Sanger Institute) as previously described[24]. As a further quality check of siRNAs, we applied a bioinformatics method GESS (genome-wide enrichment of seed sequences matches)[51]. Using settings for minimum number of one seed match, none of the siRNAs used in this study showed statistically significant seed match enrichment for off-target transcripts. As negative controls, we used mock-transfection, which includes the transfection mix without any siRNA. However, we confirmed in separate test experiments, that each of our readouts used in this study (53BP1, γ-H2AX, p53, EdU) gave comparable results when we used a non-targeting siRNA as a negative control (Ambion Silencer Select Negative control #1, Cat. No: 4390843). For siRNA transfections in 96-well plates, 25 µl of siRNA (40 nM, diluted in RNASe-free water) were mixed and incubated for 15 min with 25 µl of OPTIMEM medium (Thermo Fisher Scientific, 31985070) containing transfection reagent (1:32 dilution, HiPerfect, Qiagen, 301705) in V-bottom 96-well plates (Kisker, G060). Liquid handling was carried out with a multi-well pipetting device (Liquidator, Mettler-Toledo). The transfection mix was transferred into 96-well imaging plates (Greiner 655090) into which 150 µl of a cell suspension in culture medium was added into each well using an automated cell-seeding device (Multidrop Reagent dispenser, Eppendorf). This yields a final concentration of 5 nM siRNA per well. For imaging in 384-well plates (BD Falcon, 353962), 20 µl of the transfection mix was used with 60 µl of a cell suspension. For transfection of naïve U-2-OS cells, 4,000 cells per well were used in 96-well plates. For transfections of fluorescently tagged cell lines for live cell imaging, we used 6,000 cells per well in 96-well plates and 1,500 cells per well in 384-well plates. Transfections for live cell imaging were carried out directly in a $CO_2$-independent culture medium.

**Immunofluorescence and antibodies for IF, EdU and BrdU.** Cells grown on glass coverslips (12 mm wide, German glass #1.5) or in 96-well plates were washed once in PBS and fixed for 12–15 min with 3% ultrapure Formaldehyde in PBS. For experiments in Fig. 4a,b, cells were cultured on Fibronectin-coated coverslips (Neuvitro, GG-12-1.5-fibronectin) to minimize loss of loosely attached mitotic cells. Cells were permeabilized for 5 min using 0.2% Triton-X-100 in PBS. Primary antibodies were diluted in culture medium DMEM with 10% FBS at the following dilutions: rabbit 53BP1 (1:500, Santa Cruz, sc-22760), mouse γH2AX (1:500, BioLegend, 613401), rabbit p53 (1:500, Cell Signaling #2527), rabbit Lamin A (1:1,000, Abam, ab26300), mouse BrdU (1:500, GE Healthcare, RPN202). Secondary antibodies conjugated with Alexa Fluor dyes were diluted in culture medium (highly cross-absorbed antibodies, 1:500, Thermo Fisher Scientific) using the following products: A11034 goat anti-rabbit Alexa Fluor 488, A11029 goat anti-mouse Alexa Fluor 488, A11036 goat anti-rabbit Alexa Fluor 568, A11031 goat anti-mouse Alexa Fluor 568, A21245 goat anti-rabbit Alexa 647. Nuclei were stained with DAPI dye (0.5 µg ml$^{-1}$, D1306, Thermo Fisher Scientific) for 10 min following immunostaining. For detection of EdU, cells were incubated with 10 µM EdU 30 min before fixation and EdU detection was performed according to manufacturers recommendations (Click-iT EdU Alexa Fluor 647 imaging kit, Thermo Fisher Scientific, C10340) before incubation with primary antibodies. For detection of BrdU (Bromodeoxyuridine, Sigma, B9286) in the experiment shown in Fig. 5g, cells were labelled for 5 h with 10 µM BrdU before fixation. BrdU incorporation was detected after denaturation of DNA by a 15 min treatment with 1.5 N HCl, followed by several washes with PBS and antibody detection of BrdU.

**Western blotting and antibodies for western blotting.** For immunoblotting, cells were grown in 60 mm cell culture dishes and whole cell extracts were obtained by lysis in radioimmunoprecipitation assay (RIPA) buffer (50 mM Tris–HCl (pH 8.0), 150 mM NaCl, 1.0% Igepal CA-630, 0.1% SDS, 0.1% Na-deoxycholic acid, supplemented with protease and phosphatase inhibitors) containing Benzonase (Novagen) and analysed by SDS–polyacrylamide gel electrophoresis following standard procedures. Primary antibodies were incubated over night at 4 °C in PBS-Tween containing 5% powder milk. Secondary peroxidase-coupled antibodies (Vector labs) were incubated at room temperature for 1 h. ECL-based chemiluminescene was detected with an Odyssee-Fc system.

Antibodies used for WB are: mouseTubulin (1:1,000, Santa Cruz, sc-8035), mouse MCM2 (1:500, Novus Biologicals, H00004171-M01), rabbit CDC27 (1:1,000, Abcam, ab129085), rabbit KIF23(MKLP1) (1:1,000, Abcam, ab174304), mouse PLK1 (1:1,000, Abcam, ab17056), rabbit CENPE (1:1,000, Abcam, ab124733), rabbit ANLN (1:1,000, Abcam ab99352) and rabbit CDCA5 (1:1,000, Abcam ab192237).

**High-content microscopy.** All images for experiments carried out on multi-well plates for both live and fixed cells were acquired with the fully automated ImageXpress Micro-XL wide-field microscope (Molecular Devices) using a ×20, NA 0.75 Plan Apo objective (Nikon). Large-field images (700 × 700 µM at ×20 magnification) typically containing up to 500 cells per image in control wells were captured by a CMOS camera at 16-bit depth with a spatial resolution of 320 nm per pixel. Exposure times to fluorescent excitation light (solid state LED light source) were matched to the available fluorescent signal of the sample to avoid saturation of signals and potential phototoxicity, respectively (typically between 40 and 200 ms). Filter cubes for excitation and emission light (Semrock BrightLine series) were changed automatically to acquire the desired wavelengths. Filters installed on the microscope were chosen to maximally separate colour channels when

**Figure 5 | Failed cytokinesis triggers RS accompanied by progressive DNA breakage.** (**a**) A schematic depiction of cellular paths to ploidy increase after failed cytokinesis. (**b**) Representative image galleries (n = 10) of U-2-OS cells stably expressing mRFP-53BP1 treated with the indicated siRNAs and analysed by time-lapse microscopy. Scale bar, 10 µm. (**c**) Representative image galleries (n = 10) of U-2-OS cells stably expressing GFP-PCNA treated with the indicated siRNAs and followed by time-lapse microscopy. Numbers indicate time (h) after siRNA transfections. Arrowheads mark a cell undergoing re-replication. Scale bar, 10 µm. (**d**) Representative image galleries (n = 10) of U-2-OS cells treated and analysed as in (**c**). Numbers indicate time (h) after siRNA transfections. Arrowheads mark nuclei undergoing asymmetric DNA replication. Scale bars, 10 µm. (**e**) Representative images (n = 100) of U-2-OS cells treated with the indicated siRNA and immunostained for 53BP1. Nuclear DNA was counterstained by DAPI. Scale bar, 10 µm. (**f**) Representative images (n = 100) of U-2-OS cells treated as in (**e**) and immunostained for γ-H2AX. Scale bar, 10 µm. (**g**) Representative images (n = 25) of U-2-OS cells treated with the indicated siRNA and pre-incubated with BrdU for 5 h before fixation and immunostaining for 53BP1. Insets show ×2 magnification of the indicated areas. Scale bar, 10 µm. (**h**) Representative images (n = 25) of U-2-OS cells treated with the indicated siRNA and immunostained for 53BP1 and γ-H2AX. Nuclear DNA was counterstained by DAPI. Arrowheads mark the fragmented nuclei that lost 53BP1 and where DSBs are decorated solely by γ-H2AX. Scale bar, 10 µm. (**i**) Representative images (n = 25) of U-2-OS cells stably expressing Gam-GFP treated with the indicated siRNA and immunostained for Lamin A and γ-H2AX. Nuclear DNA was stained by DAPI. Insets show 2x magnification of the indicated areas. Scale bar, 10 µm.

combining up to four wavelengths in one experiment (DAPI filter for DAPI dye, GFP filters for GFP and Alexa Fluor 488 dye, TRITC filter for Alexa Fluor 568 dye when used in combination with CY5, otherwise TR filter, TR filter for mRFP, CY5 filter for Alexa Fluor 647 dyes). An infrared laser-based hardware autofocus function was deployed before acquisition of each field. For live cell experiments, the temperature in the microscopy was adjusted to 37 °C and plates were sealed to avoid evaporation of medium during prolonged imaging. Images were acquired at 30 min intervals over 72 h, starting 16 h after siRNA transfection. Image files and their associated quantitative measurements (MetaSeries TIF) were managed in a database included with the microscope (MS-SQL, Molecular Devices).

**Other wide-field automated microscopy.** Images in Fig. 4a,b were acquired with a ScanR inverted microscope (Olympus) using a × 20, 0.75 NA (UPLSAPO × 20) dry objective in an automated fashion. Images were processed and analysed using the propriety ScanR analysis software (Olympus, 2.6.1).

**Confocal microscopy.** Confocal images shown in Fig. 5e–i were acquired using the UltraVIEW Vox spinning disk microscope (Perkin Elmer) and Volocity software (version 6.3) using a × 60, 1.4 NA Plan-Apochromat oil immersion objective and appropriate excitation and emission filter sets for up to four different wavelengths. Images were captured by a Hamamatsu EMCCD 16-bit camera at a spatial resolution of 121 nm ($x$, $y$) and 250 nm in $z$ dimension and displayed as extended focus images (image planes projected into one plane). To avoid saturated intensities, laser power and exposure time were appropriately adjusted with identical settings applied within a series of experiments.

**Image analysis of fixed cells.** Images generated by high-content microscopy (data source for plots in Figs 1 and 2, Supplementary Fig. 3) were analysed using the Custom Module Editor function of the MetaXpress software (Molecular Devices, V5.1). The software offers a range of flexible segmentation tools to identify nuclei of different morphologies and objects within nuclei, such as 53BP1 and γ-H2AX foci, marking sites of DNA breakage. For detection of nuclei in fixed cell experiments (Fig. 1, Supplementary Fig. 3), a nuclear mask was created using the DAPI staining and appropriate settings for nuclear area, perimeter and intensities. The nuclear mask was then used to measure several intensity-based features for all readouts and to measure properties of DNA damage-induced foci within nuclei. Foci detection was performed using morphological top-hat filtering followed by threshold-based segmentation of objects. To achieve maximum robustness and sensitivity for each of the five readouts in this study, we extracted several features for quantification of nuclear morphology alterations, values for integrated intensity, average intensity, area, perimeter, cell length, cell breadth, and elliptical form factor were derived from DAPI-stained nuclei. For quantification of the 53BP1-mediated DDR reported by the focal accumulation of 53BP1 protein at sites of DNA DSBs, foci number, foci area and foci integrated intensities were measured. For the quantification of the γ-H2AX-mediated DDR, we measured nuclear integrated intensities, average intensities, minimum average intensities, maximum average intensities, foci count, foci area, foci integrated intensities, foci average intensities, foci minimum intensities and foci maximum intensities in all images. This number of features was necessary to capture the full profile of γ-H2AX responses ranging from focal to pan-nuclear distributions or a mixture of both. For the quantification of cellular stress responses reported by the increased nuclear levels of the p53 protein and for DNA replication reported by incorporation of the thymidine analogue EdU in DNA, values for average and integrated nuclear intensities were derived from all images. This analysis was executed on all three replicates and single-cell-based values were exported to Excel spreadsheets for data analysis by LR described in the next step. Because wide-field microscopy is limited by filter-based optics and does not allow simultaneous recording of five different readouts, measurements are derived from two independent sets of experiments (morphology, 53BP1, γ-H2AX and EdU in set #1 and p53 measurements in set #2).

**Image analysis of live cell time-lapse microscopy.** For the analysis of mitotic phenotypes based on nuclear morphology (data source for Fig. 2a,b), images from time-lapse recordings of U-2-OS cells stably expressing chromosomal marker protein Histone H2B-mCherry were used and analysed with the MetaXpress Custom module editor software tool. Here, the fluorescent-protein-derived signals are used to measure nuclear morphology as well as for the detection of cells in mitosis and cell death associated with mitotic perturbations. The same set of features as described above for DAPI-based nuclear morphology measurements was used to define these populations. For each image of the time series, the fraction of cells with aberrant nuclear morphology, fractions of mitotic cells and fraction of dying cells were aggregated to yield a composite value for morphology aberrations. This was normalized to control-treated wells for each time-point over the entire time-lapse recording. For the analysis of DDR after mitotic perturbation, numbers for 53BP1 foci in U-2-OS cells stably expressing mRFP-53BP1 were derived from images of time-lapse recordings using the same type of image segmentation for foci as described for fixed cells (top-hat filtering and intensity-based thresholding). Nuclear intensities of mRFP-53BP1 were used to detect number of nuclei in images and to derive values for number of foci per nucleus. Values for foci per interphase

cell were normalized to control-treated wells and displayed in the heat map diagram in Fig. 2. For optimal range indication, white colour-code of heat map reflects average of the population for each time series, values below average are blue (including control = 1) and values above average are red. Position of genes per siRNA in the heat map was sorted in this way: first, data were split into two groups depending on whether mitotic aberrations (threefold over control) or DNA damage (1.5-fold over control) occurs first in a time series. Remaining genes showing milder increases in either readout were grouped at the top of the heat map. Data were sorted according to the first time-point with increased DNA damage.

**Quantification of mitotic delays in time-lapse movies.** Duration of mitosis was analysed manually from at least two independent time-lapse movies. Averages of a minimum of 25 cells were calculated.

**Image analysis of nuclear morphology.** Images of live U-2-OS cells stably expressing fluorescently tagged Histone H4 (mEOS2 fluorescent protein) were acquired 72 h after transfection with the siRNA library (data source for Supplementary Fig. 1). Similar procedures for image analysis as described above to identify nuclear morphology aberrations, mitotic and dying cells were applied.

**Image analysis of mitotic γ-H2AX.** Images of cells growing on coverslips were acquired and analysed with the Olympus ScanR microscope and software. The DAPI signal was used for the generation of an intensity-threshold-based mask to identify mitotic cells as main objects. This mask was then applied to analyse pixel intensities for γ-H2AX foci in mitotic figures, from which a calculated parameter (average sum of foci intensity per all mitotic cells) was derived (Fig. 4a,b).

**Logistic regression-based machine learning.** For advanced data analysis of image-based measurements of nuclear morphology, 53BP1, γ-H2AX, EdU and p53, we applied a multiparametric machine learning-based approach using LR. LR is a type of regression analysis used for predicting the outcome of a dependent variable with a probability. LR quantifies the relation between the dependent variable and one or more independent variable. In LR one assumes a two-class classification problem for which the positive class is assigned class value 1 and the negative one as 0.

For example in case of 53BP1 feature type,

*Dependent variable ($y$)* = variable that determines whether or not the given cell belongs to positive siRNA (that is, class value $y = 1$) or negative (control) siRNA (that is, class value $y = 0$).

*Independent variable ($x$)* = variable that represents features such as foci number, foci area and foci integrated intensities represented by $x_1$, $x_2$ and $x_3$ respectively.

Classical LR model seeks to assign to a pattern $x_t$ a probability $\pi(x_t) = p(y_t = 1 | x_1, x_2, x_3)$ according to following expression

$$\log\left(\frac{p}{1-p}\right) = w_0 + w_1 * x_1 + w_2 * x_2 + w_3 * x_3$$

Where $p = p(y_t = 1 | x_1, x_2, x_3)$ that is, probability of dependent variable $y_t$ to be 1 given independent variable $x_1$, $x_2$ and $x_3$.

As it can be seen from above equation, LR models are actually linear models and optimal weights $w_0$, $w_1$, $w_2$ and $w_3$ are obtained by maximizing the training sample's likelihood with the help of maximum likelihood estimation method. LR uses Signal processing, statistics and machine learning, parallel computing and MATLAB Compiler toolbox available from Mathworks (custom code developed in Matlab R2016a version).

Cells from control-treated wells are sampled against cells treated with various siRNAs to invoke mitotic perturbations, well by well. The algorithm finds the best models based on the ten siRNAs with the strongest phenotypes within the data set, obliterating the need for pre-designed positive controls (which are difficult to fit multiparametric profiling approaches). These models are trained to identify the full range of cellular responses and thus return highly robust and sensitive scores. LR performance is evaluated by Area under receiver operative characteristic (ROC) curve. ROC analysis provides tools to select optimal models and discard suboptimal ones independently from the class distribution. The outcome of this analysis is a ranked set of models where the ten best models reflect the ten siRNAs with cellular phenotypes maximally different from control cells. This ensemble of LR models is used to calculate a probability score for every cell of a siRNA treated population. Inclusion of more models does not provide extra benefit for the analysis, as tested by DeLong test, a non-parametric approach to compare ROC curves. The training phase of LR machine learning was done on 66% of the data (Replicates #1 and #2). Leave-one-out cross-validation was used as a model validation technique, that is, leaving one replicate out as a test data, to assess how the results of this training set generalizes to an independent test data set (Replicate #3). Single-cell probability scores calculated in this way were ultimately aggregated to derive an overall probability score for the entire cell population of a well treated with a given siRNA. LR scores reflect both the magnitude and the frequency of cellular events in a large population. Following statistical values are computed over LR scores from individual cells within a particular siRNA CI lower

and CI upper = lower and upper 95% confidence interval boundaries for the mean; s.d.; interquartile range; sample size = number of cells involved in LR scoring within a particular siRNA for a particular replicate (test replicate no.); s.e.m.; Kolmogorov–Smirnov test (H) = indicates the result of the two-sided Kolmogorov–Smirnov goodness-of-fit hypothesis test for standard normal distribution; H = 1 means test reject the null hypothesis (that is, data could have come from a standard normal distribution) at the 5% significance ($\alpha = 0.05$) with P value (Kolmogorov–Smirnov test (P)). Following statistical values (effect size measures) are computed over LR scores from individual cells within a particular siRNA compared with respect to Control siRNA (that is, Neg siRNA): Mann–Whitney U-test (p) = two-sided Mann–Whitney U-test P value performed at $\alpha = 0.05$, significance level; Hedges' g: Effect size measure calculated by subtracting the means and dividing the result by the pooled s.d. Suitable for groups with different sample size, by adjusting the calculation of the pooled standard deviation with weights for the sample sizes. In the literature, usually this computation is called Cohen's d as well. Area under ROC (AUC): The AUC is a non-parametric effect size metric for binary classification problems. If the classifier is very good, the true positive rate will increase quickly and the area under the curve will be close to 1. If the classifier is no better than random guessing, the true positive rate will increase linearly with the false positive rate and the area under the curve will be around 0.5. Statistical data for all read-outs of all replicates are summarized in Supplementary Dataset 3.

**Other statistical tests.** Linear dependence between siRNA #1 and #2 was evaluated by Pearson product-moment correlation coefficient (r) with confidence level ($\alpha = 0.05$) and P values (p) for testing the hypothesis of no correlation as shown in Supplementary Fig. 2b.

**Cluster analysis.** To find phenotypic groups in the data sets, that are similar to each other, we applied k-means clustering (performed in Matlab) on siRNA #1 for the following variables: nuclear morphology and mitotic cells (mitotic pheno-types), DNA damage (53BP1 and γ-H2AX), general cellular stress response (p53) and DNA replication (EdU).

**Code availability.** Code for LR analysis is available from the corresponding authors (J.L., C.L.) upon reasonable request.

**Data availability.** Source data for all image analysis and LR analysis are available from the corresponding authors (J.L., C.L.) upon reasonable request.

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

## Acknowledgements

We thank Roland Kanaar (Erasmus University Medical Center, Rotterdam) and Susan Rosenberg (Baylor Colleague of Medicine, Houston) for reagents, and all members of the Lukas lab for helpful comments. This work was supported by the Novo Nordisk Foundation (NNF14CC0001 to J.L. and NNF12OC0002088 to C.L.), Danish Cancer Society (R72-A4436 to J.L.), the European Community 6th Framework Programme MitoCheck (LSHG-CT-2004-503464 to J.E.) and European Community 7th Framework Program MitoSys (241548 to J.E.).

## Author contributions

C.L. and J.L. conceived the study. R.S.P., C.L. and M.-B.R. performed the screens. R.S.P. performed the image analysis. G.K. performed supervised machine-learning and statistical data analysis. R.S.P. and G.K. performed k-means clustering, B.N., J.-K.H., R.P. and J.E. provided the siRNA libraries and bioinformatic tools. C.L., R.S.P., G.K., T.G. D.W.G. and J.L. analysed and interpreted the data. C.L. and J.L. wrote the manuscript.

## Additional information

**Competing financial interests:** The authors declare no competing financial interests.

**Publisher's note**: 

