## [Peer Review File · Nature Communications]

REVIEWERS' COMMENTS:

Reviewer #1 (Remarks to the Author):

I think the authors have done an outstanding job of revising their manuscript and I support their publication.

Reviewer #3 (Remarks to the Author):

I was asked to review this revised manuscript (I was not an original reviewer).

From my point of view, the authors have done an admirable job in addressing the original comments. The manuscript is very well written, the new set of analyses done at a high standard and there are a number of findings that will highly interest the field, my favourite being the observation that the depletion of a mitotic regulators such as KIF23 leads to asymmetric DNA breakage in binucleated cells. I am sure there are many more gems like these in the curated dataset associated with the manuscript.

Although this manuscript is more a starting point than an endpoint, I am enthusiastic about this study and strongly recommend publication

FINAL EVIEWERS' COMMENTS

Reviewer #1 (Remarks to the Author):

I think the authors have done an outstanding job of revising their manuscript and I support their publication.

Reviewer #3 (Remarks to the Author):

I was asked to review this revised manuscript (I was not an original reviewer).

From my point of view, the authors have done an admirable job in addressing the original comments. The manuscript is very well written, the new set of analyses done at a high standard and there are a number of findings that will highly interest the field, my favourite being the observation that the depletion of a mitotic regulators such as KIF23 leads to asymmetric DNA breakage in binucleated cells. I am sure there are many more gems like these in the curated dataset associated with the manuscript.

Although this manuscript is more a starting point than an endpoint, I am enthusiastic about this study and strongly recommend publication

OUR RESPONSE

We are extremely grateful to both Reviewers for their favorable assessment of our revised manuscript and for such as strong support for publication. It is very rare these days to receive such an enthusiastic response (Reviwer #3: 'I am sure there are many more gems in the curated dataset') and we take it as a strong encouragement that our work has addressed a timely and important issue and that it will have a positive impact on the large field of genome integrity maintenance.

Neither of the the two Reviewers requested additional experiments or textual amendments, and we therefore thank them once again for their time and for guiding us so well throughout the reviewing process.